# Application of Biodegradable Magnesium Membrane Shield Technique for Immediate Dentoalveolar Bone Regeneration

**DOI:** 10.3390/biomedicines11030744

**Published:** 2023-03-01

**Authors:** Akiva Elad, Patrick Rider, Svenja Rogge, Frank Witte, Dražen Tadić, Željka Perić Kačarević, Larissa Steigmann

**Affiliations:** 1Private Practice, Tel Aviv 6473313, Israel; 2Botiss Biomaterials GmbH, 15806 Zossen, Germany; 3Department of Prosthodontics, Geriatric Dentistry and Craniomandibular Disorders, Aßmannshauser Straße 4–6, 14197 Berlin, Germany; 4Department of Anatomy Histology and Embryology, Faculty of Dental Medicine and Health, Josip Juraj Strossmayer University of Osijek, 31000 Osijek, Croatia; 5Department of Oral Medicine, Infection, and Immunity, Division of Periodontology, Harvard School of Dental Medicine, Boston, MA 02115, USA

**Keywords:** socket preservation, ridge preservation, NOVAMag membrane, resorbable metal

## Abstract

For the first time, the clinical application of the first CE registered magnesium membrane is reported. Due to the material characteristics of magnesium metal, new treatment methodologies become possible. This has led to the development of a new technique: the magnesium membrane shield technique, used to rebuild the buccal or palatal walls of compromised extraction sockets. Four clinical cases are reported, demonstrating the handling options of this new technique for providing a successful regenerative outcome. Using the technique, immediate implant placement is possible with a provisional implant in the aesthetic zone. It can also be used for rebuilding both the buccal and palatal walls simultaneously. For instances where additional mechanical support is required, the membrane can be bent into a double layer, which additionally provides a rounder edge for interfacing with the soft tissue. In all reported clinical cases, there was a good bone tissue regeneration and soft tissue healing. In some instances, the new bone had formed a thick cortical bone visible in cone beam computed tomography (CBCT) radiographs of the regenerated sites, which is known to be remodeled in the post treatment period. Overall, the magnesium membrane shield technique is presented as an alternative treatment option for compromised extraction sockets.

## 1. Introduction

Alveolar ridge atrophy is the unavoidable consequence of tooth loss after extraction [1]. This progressive and irreversible phenomenon can give rise to esthetic, functional, and prosthodontic challenges as well as interfere with an ideal implant placement for tooth replacement. Hence, proactive surgical intervention at the time of extraction may help providing desired surgical and prosthetic outcomes. Several techniques that allow dental extraction, implantation, and even if desired provisionalization, have been described in the literature.

Immediate dentoalveolar restoration (IDR) [2] suggests using cortico-cancellous bone graft harvested from the maxillary tuberosity, which is shaped to the defect size and is inserted between the implant and the soft tissue in a flapless approach. However, this requires a secondary surgical site to harvest donor hard tissue for rebuilding the buccal cortical wall, which increases patient pain and morbidity. Additionally, manipulation of the graft must be performed quickly to maintain its vitality.

The socket shield technique (SST) suggests partial root retention on the buccal aspect during tooth extraction with subsequent immediate implant placement [3]. This technique is technically demanding and time intensive, and the feasibility of SST had yet to established as the potential clinical benefits of SST lack strong scientific evidence [4].

An alternative treatment procedure could be designed around the characteristics and biological effects of a new material. Instead of using an autologous bone or tooth that can be challenging and time consuming to be shaped to fit the defect, a bioresorbable synthetic material could enable shorter surgery time due to easier handling.

Magnesium metal has only recently been introduced as a regenerative dental material, with the first two magnesium medical devices receiving CE approval in 2021 [5,6]. As a material choice, magnesium is already well established for orthopedic and cardiovascular applications [7], but has yet to be reported on for its clinical application in regenerative dentistry. Magnesium has a combination of properties that make it unique as a regenerative dental material: synthetic, mechanically strong but malleable, and completely resorbable [8]. Additionally, magnesium metal degrades into magnesium ions that are naturally occurring in the body and are used in many important functions, such as the maintenance, growth, and regeneration of bone [9]. Mg ions are known to promote cortical bone growth via periosteal stem cells through the release of calcitonin gene-related peptide (CGRP) from sensory nerve endings in the periosteum [10].

The magnesium membrane has previously been demonstrated to have increased soft tissue adhesion. In vitro studies have shown that gingival human fibroblast cells (Primary HGF-1 cells) will adhere and form confluent layers on the magnesium membrane surface, as well as migrate over the surface following a scratch test [11]. These results were further proven in an in vivo beagle dog model, using membranes implanted into palatal defects, which demonstrated 90% wound closure after 72 h. This can be expected, as previous studies have reported on the role of Mg ions in promoting the adhesion of oral soft tissue cells to titanium substratum, and has even been demonstrated to be more beneficial than Ca ions, a known mediator of cell to substratum adhesion [12,13,14].

Once implanted, the magnesium metal begins to degrade until it is completely resorbed by the body. Therefore, there is no need to extract it during a second surgical procedure. During the degradation of the magnesium metal, the metallic structure is transformed into magnesium salts and a small volume of hydrogen gas is released [15]. The composition of the magnesium salts contain elements that are naturally present within the bone matrix, have a good biocompatibility, and can become enveloped in new bone [16]. The magnesium salts keep the original shape and position of the membrane until they are resorbed by the body, whereas the small volume of hydrogen gas initially provides a slight tenting of the soft tissue. Both of these degradation by-products continue to maintain a separation of the soft and hard tissues [5,17]. As reported for other magnesium implants, once the magnesium metal has degraded, no more hydrogen gas is released and the space created by the gas spontaneously resolves [18,19]. Magnesium implants have already been proven to provide an excellent biocompatibility in applications such as cardiovascular stents [20,21] and orthopedic screws [22,23].

Magnesium ions that are resorbed by the body during the degradation process, are already prevalent within the body and are found within almost every single cell [24]. As magnesium ions are already well established within the body, there are pathological pathways for the excretion of excessive levels of magnesium ions [24].

An in vivo performance study in Beagle dogs, demonstrated that the magnesium membrane was able to provide a similar result to that of a collagen membrane when used to augment four-wall defects in combination with bovine bone graft [17]. A similar ratio of new bone volume to soft tissue volume was found at all timepoints when comparing the magnesium membrane and collagen membrane treatment groups, demonstrating that the magnesium membrane was just as effective as the collagen membrane at maintaining a barrier and enabling the bone within the defect to regenerate. In the veterinarian report, neither the magnesium membrane nor the collagen membrane group presented signs of a chronic inflammation reaction such as prolonged redness, swelling, pain, and loss of function.

In the performance study, one week post implantation, the magnesium membrane had a slightly higher inflammatory reaction; however, this quickly subsided, and by week 8, the subsequent timepoint in the study, the inflammatory reaction was similar to that of the collagen membrane. Both membranes followed a comparable inflammatory process, involving the same type and number of immune cells [5]. A similar inflammatory response of the magnesium membrane and collagen membrane has also been reported in another in vivo study, comparing a physical vapor deposition (PVD) coated magnesium membrane, a non-coated magnesium membrane and a collagen membrane implanted subcutaneously in BALB/C mice [25]. The coated magnesium membrane induced the highest level of inflammation; however, the uncoated magnesium membrane and the collagen membrane were associated with similar levels of inflammation. Both the collagen and un-coated magnesium membrane had balanced levels of M1 and M2 macrophages, in a ratio promoting the integration of the material.

Due to the magnesium membrane’s mechanical strength [5], it also has the possibility to be used as a cortical plate. Despite being mechanically strong, the magnesium membrane can be cut and bent to shape to perfectly match the contours of the defect, potentially making it more clinically manageable than using an autogenic or allogenic bone cortical plate. Once in position, the membrane provides a barrier between soft and hard tissues and fully resorbs after the critical healing period [17]. As the degradation process is gradual, defect stability is maintained during the critical healing phase whilst new bony tissue infiltrates the defect site [5,17].

The aim of this paper is to report on the first clinical application of the magnesium metal membrane and demonstrate the application of a completely new technique. The magnesium membrane shield technique is used to rebuild the buccal wall and offers a novel clinical option due to the material properties of magnesium. The magnesium membrane is presented as an alternative material choice to the current gold-standard use of a cortico-cancellous bone plate or tooth root, with an aim to reduce invasiveness and the surgical limitations of the alternative techniques.

## 2. Materials and Methods

For each case, a minimally invasive buccal and palatal flap elevation was performed, followed by a thorough debridement of the extraction site. A periosteal releasing incision was performed on the buccal flap, 1 mm apical to the lower border of the defect. An immediate implant was placed with a high insertion torque. A sterile magnesium membrane (NOVAMag^®^ membrane, botiss biomaterials GmbH, Berlin, Germany) with an initial thickness of 140 µm and dimensions 30 × 40 mm was prepared. The membrane was trimmed to size using the NOVAMag^®^ scissors (Carl Martin GmbH, Solingen, Germany) (Figure 1A), ensuring smooth rounded edges to avoid sharp points that could perforate the soft tissue. The rim of the membrane is flattened using the NOVAMag^®^ sculptor (Carl Martin GmbH, Solingen, Germany) (Figure 1B). The membrane was then either placed as a singular layer, or bent in half into a double layer of the membrane to improve its mechanical stability. The shape of the membrane was then bent according to contours of the defect using the NOVAMag^®^ sculptor (Figure 1C). The membrane was then inserted into the socket as an extension of the buccal plate. The defect was filled with allogenic granular bone substitute (maxgraft^®^, botiss biomaterials GmbH, Berlin, Germany) and a collagen membrane (Jason^®^ membrane, botiss biomaterials GmbH, Berlin, Germany) placed over the top of the augmentation and the flaps were sutured (Figure 1D,E). Closed wound healing is recommended with the magnesium membrane, as exposure can cause an accelerated resorption time. Suturing was performed with Nylon 5-0 sutures, using single interrupted and vertical mattress suturing techniques. Then, implants were immediately loaded (Figure 1F).

Post-operatively, the patients were instructed to rinse with chlorhexidine solution twice a day for 2 weeks.

## 3. Results

### 3.1. Case 1

The patient was a 65-year-old female in good general health condition. The patient presented with tooth 24, root canal treatment, post, core build up, and an old porcelain-fused-to-metal (PFM) crown. A vertical root fracture was present with an associated severe bone loss, including loss of both buccal and palatal plates.

The non-traumatic extraction of tooth 24. was performed, (Figure 2). The magnesium membrane shield technique was performed as described in Section 2, aiming to rebuild both buccal and palatal plates. In this instance, the magnesium membrane was used as a single layer on both the buccal and palatal sides.

Four months post operatively, regenerated bone was present, including fully regenerated cortical and palatal plates (Figure 3). The implant was stable and there was a good healing of the soft tissues.

### 3.2. Case 2

The patient was a 68-year-old male in good general health condition. The patient presented with tooth 25, root canal treatment, crown destroyed at the tissue level and a vertical root fracture with associated severe bone loss including loss of buccal plate. The palatal plate remained intact.

A non-traumatic extraction of tooth 25 was performed (Figure 4), followed by bone augmentation using the magnesium membrane shield technique to rebuild the buccal wall as described in Section 2. In this instance, the magnesium membrane was bent into a double layer.

Four months post operatively, there was a good regeneration of bone within the defect including a fully regenerated cortical buccal plate (Figure 5). The implant was stable and the was a good healing of the soft tissues.

### 3.3. Case 3

A 62-year-old male patient in good general health condition. Patient presented with tooth 21, vital tooth, horizontal oblique root fracture with associated severe bone loss including loss of buccal cortical plate. Palatal cortical plate remained intact.

Nontraumatic extraction of tooth 21 (Figure 6), followed by bone augmentation using the magnesium membrane shield technique to rebuild the buccal wall as described in Section 2. The implant was immediately covered with a provisional crown due to its location in the aesthetic zone.

Four months post operatively, there was an excellent regeneration of bone defect including a fully regenerated cortical buccal plate (Figure 7). The implant was stable, and a good healing of the soft tissues was observed in the postoperative phase.

### 3.4. Case 4

A 60-year-old female patient in good general health condition. Patient presented with tooth 15, root canal treatment, post core and PFM crown and vertical root fracture with associated severe bone loss including loss of buccal cortical plate. Palatal cortical plate was intact.

Nontraumatic extraction of tooth 15 (Figure 8), followed by bone augmentation using the magnesium membrane shield technique to rebuild the buccal wall as described in Section 2.

Four months post operatively, there was an excellent regeneration of bone defect including a fully regenerated cortical buccal plate (Figure 9). The implant was stable and there was a good healing of the soft tissues.

## 4. Discussion

In regenerative dentistry, there is a wide choice of available materials that can be applied for hard tissue augmentation [26]. In order to overcome the negative consequences of tooth extraction, different techniques of alveolar ridge preservation have been proposed to retain the original ridge dimension, including immediate or delayed implant placement, and buccal overbuilding with bone grafting materials. The loss of buccal and palatal bone in periodontally compromised teeth or during extraction are a common clinical scenario that surgeons are faced with [27]. Each patient needs to be diagnosed separately and an appropriate choice of material must be chosen. Especially in the case of buccal and/ or palatal bone loss, the clinician needs to make a choice of the most suitable material, sometimes having to prioritize mechanically strong materials over resorbable or more clinically manageable materials [28].

Large size bone defects that have lost both vertical and horizontal dimensions usually require a choice of membrane or plate that will provide mechanical stability during bone regeneration of the extraction socket space. This is to counteract the external forces on the defect space created by the overlying soft tissue. The choice of materials include titanium or titanium reinforced membranes [29], collagen membranes [30], or autologous grafts [2,3].

Titanium based materials provide high mechanical stability to enable bone regeneration, which is beneficial for vertical bone gain [28]. However, as titanium is non-resorbable, it is required to be removed in a second surgery, or potentially needs a more invasive surgery upon re-entry by requiring a larger flap to be opened to access the membrane.

Collagen membranes are the most frequently used membranes in regenerative dentistry and have an excellent biocompatibility [31], but are often unstable and can deform or even collapse under loading with the soft tissue entering the bone defect [32,33].

Autologous cortical plates are mechanical strong and have an excellent biocompatibility, but require a second surgical site, increasing patient pain and morbidity. Whilst the use of a tooth root, as in SST, is technically demanding and labor intensive [34].

The development of a magnesium membrane provides a material option that is resorbable yet offers mechanical stability during the critical healing phase [5,17]. Magnesium metal releases magnesium ions during degradation that are naturally present in the body and play an essential role in many important processes within the body [35]. This includes aspects of bone regeneration such as bone cell proliferation, migration, and alkaline phosphatase activity [9]. As the magnesium metal is malleable, it potentially provides an easier adaption to the contours of the defect than is possible with a cortical bone graft. Even though the membrane is malleable, the handling of the membrane must be taken into consideration. The membrane needs to be shaped and cut during surgery in a procedure that is more technically demanding than the application of a collagen membrane and is more comparable to that of a titanium mesh. To aid with the procedure, specialized tools (e.g., NOVAMag^®^ scissor, NOVAMag^®^ sculptor) have been developed for cutting and bending the membrane to shape, reducing the risk of sharp edges that could rupture the mucosa.

Titanium meshes provide mechanical stiffness to support the defect, but do not provide a barrier function for bone regeneration [28]. From a biophysiological perspective, it is suggested that for larger sized defects, maintaining regenerative space and having good osteogenic properties is more beneficial than maintaining a cell occlusivity between the soft and hard tissues [28]. In contrast, the magnesium membrane maintains its mechanical strength during the initial critical healing period [5], but has also been demonstrated in in vivo studies to provide a stable barrier between the soft and the hard tissue [17]. Additionally, as the magnesium metal degrades and is transformed into magnesium salts that are resorbed, nutrient flow into the defect space is possible. Therefore, there is no need to perforate the membrane with holes, which could otherwise negatively affect the degradation rate of the membrane and compromise the soft tissue barrier.

In addition, it has been shown in a previous study by Rider et al. that during the degradation of the magnesium metal, hydrogen gas is released, which has been shown to provide a tenting of the soft tissue [17]. This provides an additional barrier to separate the soft from the hard tissues during the critical healing period.

Alternatively, autogenic or allogenic cortical plate grafts can be used to provide a stable mechanical support [2]; however, these too have their disadvantages. Autogenic grafts require a second surgical site and are associated with donor site morbidity. Additionally, surgical times are prolonged, which can increase the risk of infection. Both autogenic and allogenic cortical plate grafts must be shaped during surgery, which can be technically demanding. In SST [3], partial root retention is used to support the buccal plate and avoid negative tissue alteration post extraction, in a procedure that is technically demanding and time consuming. Magnesium membranes offer a synthetic alternative, as shown by the cases provided in this study, to restore the buccal and the palatal plate during extraction healing with immediate implant placement and even placing provisional restauration if indicated.

The four clinical cases demonstrate the use of a magnesium membrane to provide the necessary stability for bone regeneration and integration of the dental implant.

In each instance, the magnesium membrane was handled slightly differently, but the principle of the magnesium shield technique remained the same. In the first case, the specific requirements of the defect meant that both the buccal and palatal walls had to be built using the magnesium membrane shield technique. Used as a single layer and placed inside the periosteum with the addition of a collagen membrane over the top of the ridge, enough mechanical support was provided on the two opposing sides of the defect to enable regeneration of the bone and support the implant. Potentially this method could be used to treat a much larger sized defect and improve the vertical bone gain.

In the second and fourth case, the membrane was bent into a double layer to provide additional mechanical strength to support the defect space. By bending the membrane into a double layer, a rounded edge is produced that is better for interacting with the soft tissue. If needed, the membrane could also be secured in place using fixation screws in a similar application to the Khoury technique, replacing the need to harvest an autologous cortical bone plate.

In the third case, the membrane was placed in a single layer on the buccal side and the implant was immediately treated with provisional restauration due to its location in the aesthetic zone. A major benefit of using the magnesium membrane shield technique in the aesthetic zone is that there is no need for a large flap created during the augmentation surgery. At 4 months, there was a good implant stability and soft tissue healing.

Due to the handling properties of the membrane, it was easily adapted to the contours of the defect during surgery. All patients had a good recovery, and the healing of the sites was uneventful. In the CBCT images taken 4 months post implantation, cortical bone is visible on the newly formed trabecular bone, indicating a high quality of bone.

In each of the described cases, the magnesium membrane was held in position by being placed under the periosteum. However, depending on the indication, it might be necessary to secure the membrane in place using a fixation system. The membrane has reportedly been secured using titanium fixation screws [5,17], and can also be secured with magnesium alloy fixation screws [6,36], thereby creating a regenerative approach using only fully resorbable materials.

The use of the magnesium membrane shield technique to treat compromised extraction sockets has demonstrated the potential implantation possibilities that have been created by the development of the magnesium membrane. It provides a new material choice that is synthetic, mechanically strong, promoting cortical bone formation, malleable and completely resorbable, a unique combination of properties for regenerative dentistry. These properties are especially beneficial for supporting the defect space and the final regenerative result.

## 5. Conclusions

For the first time, a magnesium membrane has been reported on for its clinical application in regenerative dentistry. Due to its unique material properties, a new technique is possible: the magnesium membrane shield technique, whereby a magnesium membrane is used to rebuild the buccal or oral walls in compromised sockets. There are already several techniques available for rebuilding or preserving the alveolar ridge post-tooth extraction, but in comparison to the current material choices, magnesium has many beneficial properties: it is completely resorbable and does not need to be extracted; it is synthetic and therefore does not need to be sourced from an additional surgical site, as is the case for autologous cortical bone; it promotes cortical bone growth; it is mechanically strong and able to stabilize the defect space; it is malleable and can be shaped to the contours of the defect in a less technically demanding process than IDS or SST. Four clinical cases demonstrating the use of the magnesium membrane shield technique have shown excellent bone tissue regeneration and, in some instances, the new bone has formed a thick cortical bone layer.

## Figures and Tables

**Figure 1 biomedicines-11-00744-f001:**
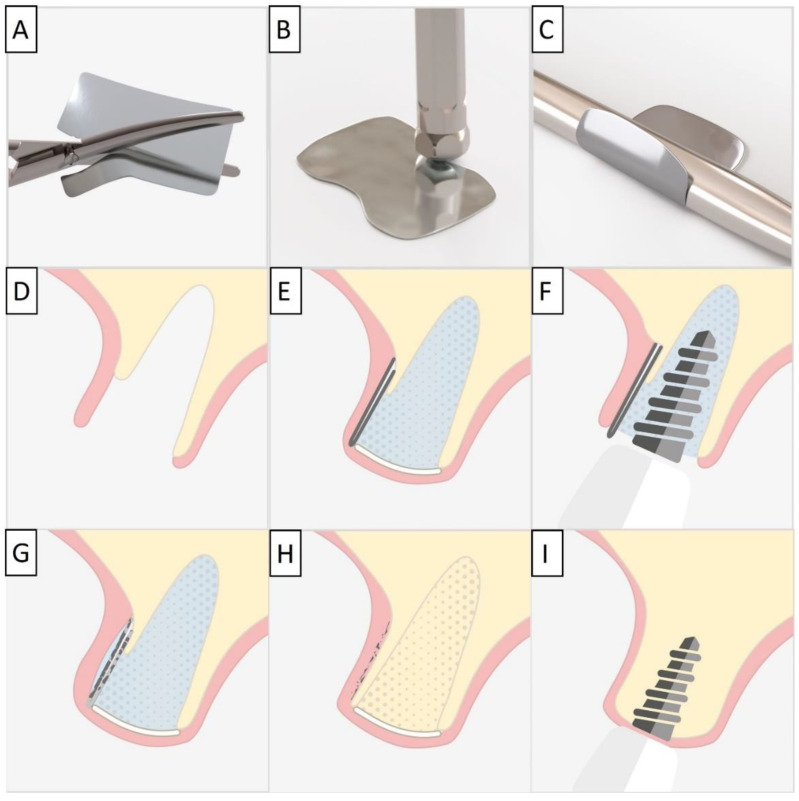
For the magnesium shield technique, the magnesium membrane is first cut to shape using the NOVAMag^®^ scissors (**A**). The rim of the membrane is then flattened using the NOVAMag^®^ sculpture (**B**) and bent into shape (**C**). In a compromised extraction socket (**D**), the membrane is either place as a single layer or bent into a double layer, before being positioned to rebuild either the buccal or palatal wall. The membrane is held in position by the periosteum and the defect space is filled with the graft material. A collagen membrane is placed on the top of the ridge (**E**). Using the technique, it is also possible to immediately place implants with a provisional restoration (**F**). Once implanted, the magnesium membrane will begin to degrade, transforming into magnesium salts that maintain the soft tissue barrier, and hydrogen gas will be released that provides a tenting of the soft tissue, which also extends the barrier effect, since cells cannot cross the gas cavity (**G**). After the magnesium metal has transformed into magnesium salts, no more gas is released, and the soft tissue returns into position over the newly formed bone and magnesium salts (**H**). After the critical healing period, the magnesium membrane is completely resorbed (**I**).

**Figure 2 biomedicines-11-00744-f002:**
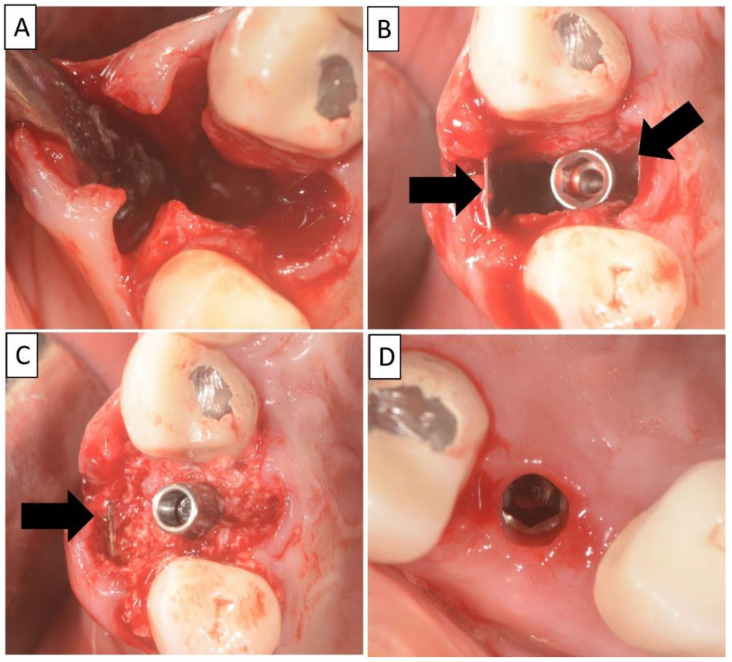
(**A**) Alveolar socket following atraumatic extraction and curettage. Severe bone loss of both buccal and palatal plates. (**B**) Buccal and palatial plates were created using the magnesium membrane shield technique. (**C**) Application of allograft. (**D**) Four months post operatively there was an excellent regeneration of bone defect, including fully regenerated cortical and palatal plates. The implant was stable and there was a good healing of the soft tissues. Black arrows are used to indicate the position of the magnesium membrane.

**Figure 3 biomedicines-11-00744-f003:**
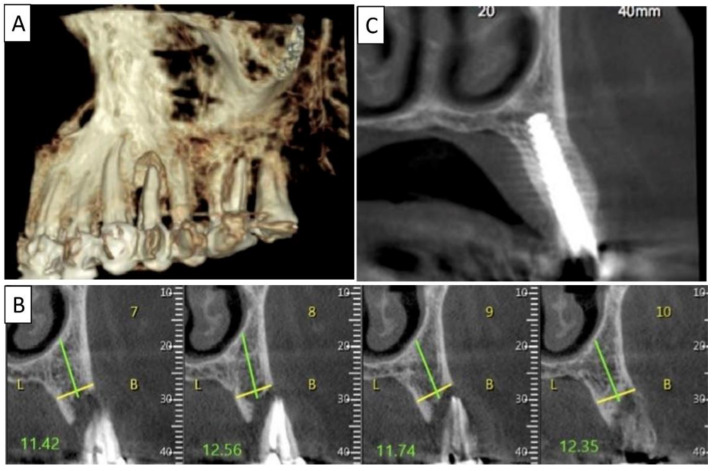
(**A**) Lateral volumetric cone beam computed tomography (CBCT) shows significant loss of buccal and palatal bone mass around tooth 24. (**B**) Coronal CBCT slice of the same area of tooth 24 revealing missed buccal and palatal bone and associated apical radiolucency. (**C**) The coronal CBCT section shows the implant in the area of tooth 24 and the obtained completely regenerated cortical and palatal plates.

**Figure 4 biomedicines-11-00744-f004:**
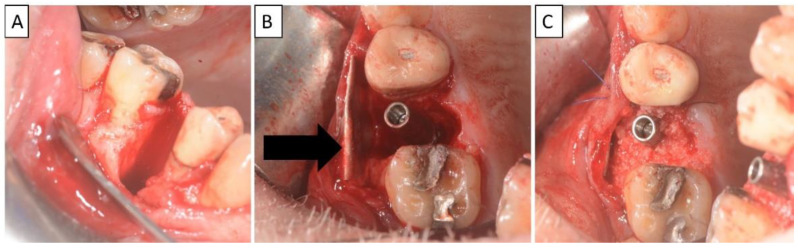
(**A**) Alveolar socket following atraumatic extraction and curettage. Severe bone loss of buccal plate. (**B**) Buccal plate was created using the magnesium membrane shield technique. (**C**) Application of allograft together with magnesium membrane. The black arrow indicates the position of the magnesium membrane.

**Figure 5 biomedicines-11-00744-f005:**
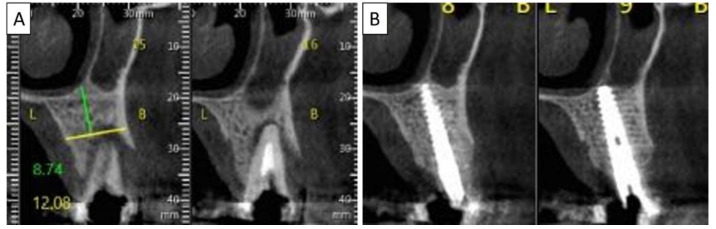
(**A**) Coronal CBCT section shows tooth 25 with vertical root fracture and destroyed buccal bony wall and intact palatine. (**B**) The coronal CBCT section shows the placed implant in the region of tooth 25 and the complete regeneration of the cortical wall.

**Figure 6 biomedicines-11-00744-f006:**
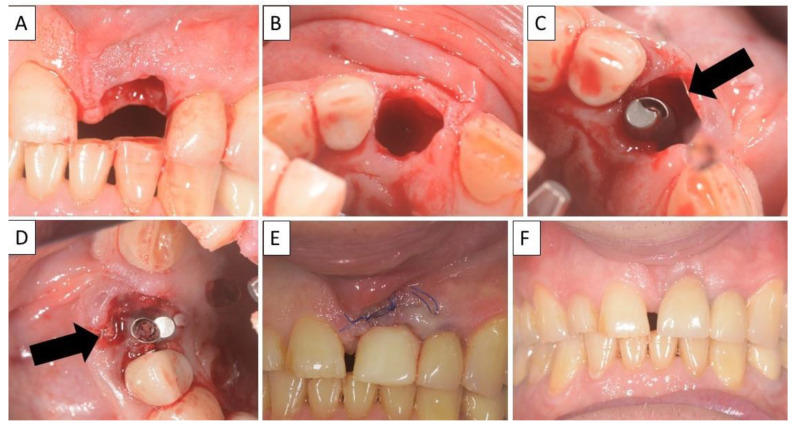
(**A**,**B**) Alveolar socket following atraumatic extraction. Severe bone loss of buccal plate. (**C**) Buccal plate was created using the magnesium membrane double layer technique. (**D**) Application of allograft. (**E**) closing sutures and immediate provision. (**F**) four months postoperatively there was an excellent regeneration of bone defect, including fully regenerated cortical and palatal plates. The implant was stable and there was a good healing of the soft tissues. Black arrows are used to indicate the position of the magnesium membrane.

**Figure 7 biomedicines-11-00744-f007:**
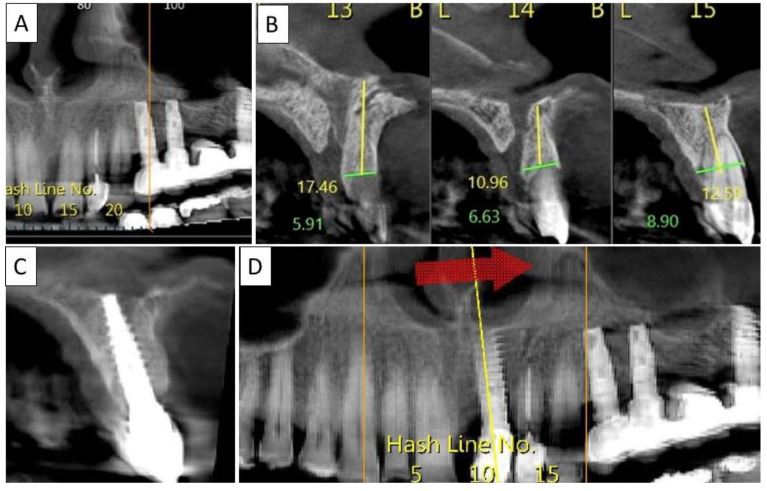
(**A**) Panoramic CBCT section shows bone loss around tooth 21. (**B**) Sagittal CBCT section of the area around tooth 21 shows extensive buccal bone loss. (**C**) A sagittal CBCT section shows the implant placed at site 21 with a restored cortical buccal plate. (**D**) Panoramic CBCT section shows bone formation around the implant in area 21.

**Figure 8 biomedicines-11-00744-f008:**
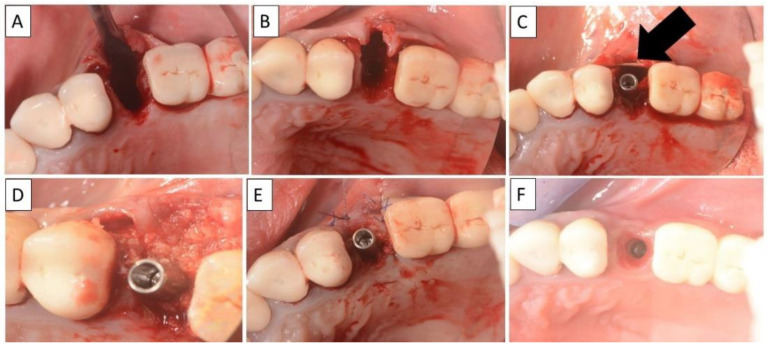
(**A**,**B**) Alveolar socket following atraumatic extraction and curettage. Severe bone loss on buccal wall. (**C**) Buccal wall was created using the magnesium membrane shield technique. (**D**) Application of allograft. (**E**) Closing sutures and immediate provision. (**F**) Four months post operatively there was a regeneration of bone defect, including fully regenerated cortical bone. The implant was stable and there was a good healing of the soft tissues. The black arrow indicates the position of the magnesium membrane.

**Figure 9 biomedicines-11-00744-f009:**
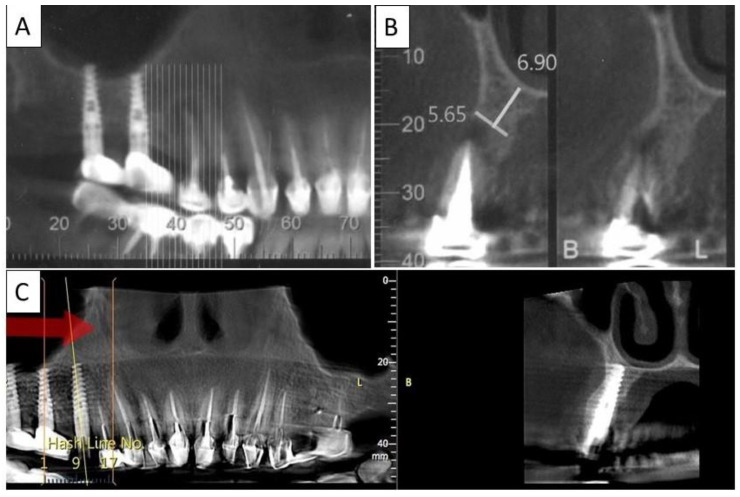
(**A**) Panoramic CBCT section shows associated apical radiolucency of tooth 15. (**B**) Coronal CBCT section shows tooth 15 with vertical root fracture and destroyed buccal bony wall and intact palatine. (**C**) Panoramic CBCT section shows bone formation around the implant in area 15. The coronal CBCT section shows the implant placed in the region of tooth 15.

## Data Availability

Data sharing is not applicable to this article.

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
