# Peer review of "Application of Biodegradable Magnesium Membrane Shield Technique for Immediate Dentoalveolar Bone Regeneration"

_biomedicines, 2023, doi:10.3390/biomedicines11030744_

Round 1
Reviewer 1 Report
This is an interesting paper that demonstrates the application of a new technique – the magnesium membrane shield technique in rebuilding the buccal or palatal walls of compromised extraction sockets.
1. In the abstract the authors have presented their noteworthy research with one or two acronyms: these terms need to be explained before report the single acronyms.
2. In the Introductory section in line 67, CGRP acronym need to be explained.
3. Quotation should be put at the end of the sentence in the line 67.
4. In the line 82, Fig 1.G was cited. It should be removed, since Figure 1 is presented in the Materials and Methods section.
5. In the Results section in line 163 it is written that magnesium membrane shield technique is described in Section 4. What is Section 4? It should be replaced with Figure 1 throughout the text.
6. In the Discussion section in line 337, it is written: “cortical bone is visible on the newly developed bone, indicating…”. This should be replaced with: “cortical bone is visible on the newly formed trabecular bone, indicating…..”
7. Ethical approval certificate number should be added.
Reviewer 2 Report
Elad et al. presented a manuscript entitled: "Application of biodegradable magnesium membrane shield technique for immediate dentoalveolar bone regeneration", where the authors showed a new technique. The figures are really well made and facilitate the understanding of the overall manuscript. The merge of industry and academia is a positive characteristic of the manuscript, highlighting the application of an idea in real patients. However, the language sometimes is not too scientific sounded. A general improvement is suggested. In general, the manuscript is a nice addition to the special issue. A few specific suggestions are listed below:
1. The title in the journal system has a misspelling: “Application of biodegradablemagnesium membrane shield technique for immediate dentoalveolar bone regeneration”. Please make sure there are no issues in the final version.
2. Lines 61-63. Please add a reference to the statement about magnesium.
3. Lines 68-73. Please introduce other studies that studied magnesium and soft tissue adhesion.
4. Line 79. Please deepen the discussion about magnesium biocompatibility. Is there really no toxicity involved? Additional referencing can improve the fact.
5. Lines 88-94. Did this study discuss biocompatibility?
6. Line 101. Similarly, introduce the in-vivo study of reference 14.
7. Fig 3. The term CBCT has not been introduced until now, please use the full name.
8. Is there any Ethics Study number/certificate associated with this manuscript?
